# Design of efficient generalized digital fractional order differentiators using an improved whale optimization algorithm

Mohammed Ali Mohammed Moqbel[1,2], Talal Ahmed Ali Ali[1,2,3], Zhu Xiao[1,2] and Amani Ali Ahmed Ali[3]

[1] College of Computer Science and Electronic Engineering, Hunan University, Changsha, China
[2] Shenzhen Research Institute, Hunan University, Shenzhen, China
[3] Taiz University, Taiz, Yemen



## ABSTRACT

This article proposes a new design and realization method for generalized digital fractional-order differentiator (GFOD) based on a composite structure of infinite impulse response (IIR) subfilters. The proposed method utilizes an improved whale optimization algorithm (IWOA) to compute the optimal coefficients of IIR subfilters of the realization structure. IWOA is developed by incorporating a piecewise linear chaotic mapping (PWLCM) and an adaptive inertia weight based on the hyperbolic tangent function (AIWHT) into the framework of original whale optimization algorithm (WOA). Simulation experiments are conducted to compare the performance of our method with that of well-known techniques, real-coded genetic algorithm (RCGA), particle swarm optimization (PSO), and original WOA. The results show that the new metaheuristic is superior to the other metaheuristics in terms of attaining the most accurate GFOD approximation. Moreover, the proposed IIR-based GFOD is compared with state-of-the-art GFOD, and observed to save about 50% of implementation complexity. Therefore, our method can be utilized in real-world digital signal processing applications.

## INTRODUCTION

Fractional derivative has gradually superseded conventional derivative in many engineering disciplines, *e.g.*, signal processing, control, and biomedical applications (*Krishna, 2011*; *Nayak et al., 2019*; *Krishna, 2019*; *Habeb et al., 2024*). This indeed owing to its ability to model the dynamic behavior of physical systems more accurately (*Xue, 2017*). Based on the Laplace transform of the Riemann–Liouville fractional derivative under zero initial conditions for order $p$, $0 < p \in \mathbb{R} < 1$, fractional differentiator is described by $s^p$ (*Vinagre, Chen & Petráš, 2003*). In this article, we deal with the problem of implementing $s^p$ by means of digital filters to compute the fractional derivative of discrete-time signals. Such a problem has long been recognized as an approximation problem, where the frequency response of $s^p$ is approximated by that of the digital filter. The frequency response of $s^p$ is obtained by substituting $s \rightarrow (j\omega)$,

Corresponding authors
Talal Ahmed Ali Ali,
taaw2012@hnu.edu.cn
Zhu Xiao, zhxiao@hnu.edu.cn

$$F_c(\omega) = (j\omega)^p = |\omega|^p e^{j\mathrm{sgn}(\omega)p}, \quad |\omega| < \pi, \tag{1}$$

where $\omega$ is the angular frequency in rad/sec and $\mathrm{sgn}(\omega)$ is the signum function,

$$\mathrm{sgn}(\omega) = \begin{cases} -1 & \omega < 0, \\ +1 & \omega \geq 0. \end{cases} \tag{2}$$

Note that $|\omega|^p$ is the magnitude response of $F_c(\omega)$ and $\mathrm{sgn}(\omega)p$ is the phase response of $F_c(\omega)$. Basically, the digital approximation can be either finite impulse response (FIR) or infinite impulse response (IIR) filter, where the frequency response is given by substituting $z \to e^{j\omega}$ in the transfer function,

$$F(z) = \sum_{n=0}^{L} b(n)z^{-n}, \tag{3}$$

$$F(z) = \frac{\sum_{n=0}^{L} b(n)z^{-n}}{1 + \sum_{n=1}^{L} a(n)z^{-n}}, \tag{4}$$

where $L$ is the filter order. Thus, the design problem is to compute either the coefficients $\{b(n)\}_{n=0}^{L}$ (for FIR) or the coefficients $\{b(n)\}_{n=0}^{L}$ and $\{a(n)\}_{n=1}^{L}$ (for IIR) such that $F(e^{j\omega})$ follows $F_c(\omega)$ as close as possible. The realization of these structures is shown in Fig. 1. While FIR filters are non-recursive with inherent stability, and can achieve exact linear phase response, IIR ones leverage recursive structures, introducing conditional stability and nonlinear phase responses. However, IIR achieve computational efficiency with lower latency through recursive designs, making them attractive for resource-constrained and delay-critical applications.

The simplest and most common version of this problem is to design a digital filter that approximates $F_c(\omega)$ with fixed fractional order $p$. Here, the magnitude and phase responses of the digital filter approximation, which we refer to as fixed fractional order differentiator (FFOD), are fixed post-design, corresponding only to an individual value of $p$. Two well known design methods for FFODs have been proposed in the literature. The first method is based on discretization methods with direct (*Visweswaran, Varshney & Gupta, 2011*; *El-Khazali, 2014*; *Rajasekhar & Krishna, 2018*; *Vasi et al., 2019*; *Rajasekhar & Krishna, 2020*; *Rajasekhar, 2022*) and indirect (*Krishna, 2011*; *Maione, 2013*; *Krishna, 2019*; *Ali et al., 2024*) schema. The other method involves formulating the approximation problem as an optimization problem and solving it using an iterative procedure. FFODs have been computed using gradient (*Johansson, 2013*) and metaheuristic (*Moqbel, Ali & Xiao, 2024*) techniques.

A hard version of the problem is to design a digital filter approximating $F_c(\omega)$ with variable fractional order $p$; we refer to this approximation as variable fractional order differentiator (VFOD). In contrast to FFOD that simply utilizes FIR or IIR structure, VFOD employs tunable composite structure (*e.g.*, Farrow structure), enabling on-the-fly reconfiguration of fractional order of derivative without redesigning the entire system. This adaptability makes VFODs ideal for applications where real-time adjustment of $p$ is necessary, *e.g.*, estimation of $1/f$ noise and handwritten signature verification. Nevertheless, VFODs often incur higher computational complexity and resource overhead

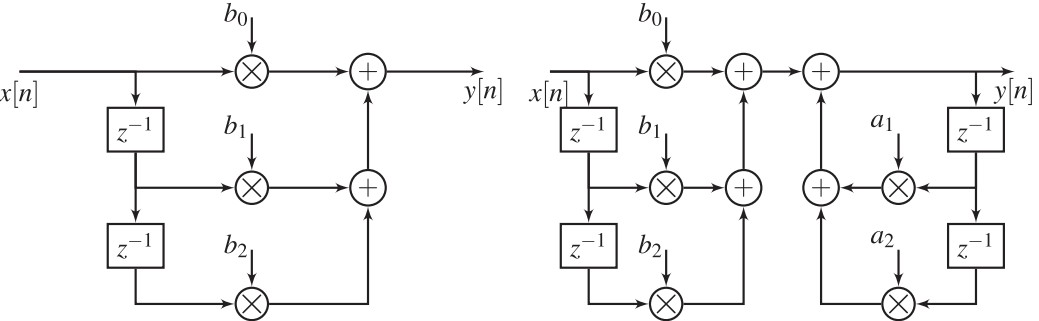

**Figure 1 Digital filter structures, FIR structure (left hand) and IIR structure (right hand).**

compared to their FFODs counterparts, as their flexibility demands additional arithmetic operations or memory for parameter storage. A yet more complex version of the problem involves the design of generalized fractional order differentiator (GFOD), which is a digital filter that approximates

$$H_d(j\omega) = |\omega|^p e^{j \, \text{sgn}(\omega)\theta}, \tag{5}$$

with $p$ and $\theta$ are variable. By utilizing two adjustable parameters, GFOD offers more flexibility than VFOD does, where the magnitude and phase responses can be independently controlled. The benefits of GFODs have been demonstrated in generating a secure single side band (SSB) signal for saving the transmission bandwidth, and in edge detection application.

Despite their potential for a variety of applications, VFODs and GFODs, in contrast to FFODs, have not received much attention from scholars. In general, VFOD has been realized using a composite structure of FIR subfilters whose coefficients are computed based on series expansion (*Tseng, 2008*), Park-McClellan minimax method (*Chan, Shyu & Yang, 2010*), or weighted least squares method (*Tseng, 2004, 2006; Shyu, Pei & Chan, 2009; Tseng & Lee, 2011a, 2011b, 2012a, 2012b*). The realization of GFOD has been carried out using a composite structure of FIR subfilters whose coefficients are optimized using weighted least squares method (*Tseng & Lee, 2015*). Nevertheless, existing approximations mainly rely on FIR subfilters, requiring high computational complexity and high latency. Therefore, it remains a real design challenge to balance adaptability with efficiency and/or latency in dynamic systems, especially for resource-constrained and delay-critical applications.

To address this challenge, we attempt to design an efficient GFOD using a composite structure whose subfilters are realized using IIR structures rather than FIR ones. This attempt is motivated by the computational advantages of IIR filters over FIR filters. First, to satisfy given design specifications, IIR filters require much fewer coefficients than FIR filters (often 1/6 of those of their FIR counterparts). Second, their group delay is much smaller than that of FIR ones, resulting in lower system latency. These merits, however, come at the cost of a significantly more challenging design problem. This is owing to the presence of the denominator polynomial in their transfer function, which renders the

formulated optimization problem highly non-convex and difficult to solve, necessitating very sophisticated iterative procedures. Yet, to attain stable transfer functions, stability constraints must be explicitly imposed during design process, forcing the poles to lie within the unit circle of the $z$ plane. To tackle these design challenges, we introduce a novel optimization technique by improving a recently-proposed metaheuristic method, whale optimization algorithm (WOA). WOA, which is inspired by the hunting strategies of humpback whales (*Mirjalili & Lewis, 2016*), has attracted many of research interests in solving real-world application optimization problems (*Nadimi-Shahraki et al., 2023*), including FFOD design problem (*Nayak & Kumar, 2024*). The attractive features of WOA are its simplicity and powerful search capability. However, like other metaheuristic algorithms, WOA faces challenges such as falling into the local optima, premature convergence, and low population diversity. As a consequence, many WOA variants have been proposed recently (*Nadimi-Shahraki et al., 2023*). Unlike previous variants, we overcome these challenges by integrating a chaotic mapping and an adaptive inertia weight into original WOA framework, facilitating the search for global optimal solutions of the highly non-convex design problems of IIR subfilters.

The contributions made in this article are manifolded.

(1) We propose an improved version of WOA (IWOA) by integrating a piecewise linear chaotic mapping (PWLCM) and an adaptive inertia weight based on the tangent hyperbolic function (AIWHT) into original WOA framework. PWLCM is adopted to provide an even population distribution and to increase population diversity. Whereas, AIWHT is utilized to mitigate local optima stagnation and to promote a fast convergence to the global optimal solution.

(2) We propose a new efficient design and realization of GFOD based on a composite structure of IIR subfilters. To solve the highly non-convex design problems of IIR subfilters, the proposed IWOA is employed, computing the optimal coefficients of the subfilters. We consider the design and realization of GFOD with fixed magnitude response and variable phase response and those of GFOD with variable magnitude response and variable phase response.

(3) We compare the performance of the proposed IWOA with that of the well-known metaheuristics, real-coded genetic algorithm (RCGA), particle swarm optimization (PSO), and original WOA, through two design examples. The proposed GFODs are also compared with state-of-the-art designs in terms of accuracy and efficiency.

(4) We experimentally verify the effectiveness of the proposed GFOD in edge detection of a square pulse, underscoring its potential and necessity for digital signal processing applications.

The remaining of this article is organized as follows. "The Design of GFODS" presents the new design of GFODs, where "Problem Formulation and Realization Structures" describes the formulation of the design problems and the realization structures of GFODs, and "Optimization Algorithm" illustrates the proposed optimization method for computing efficient GFODs. Design examples along with comparisons with existing

designs are presented in "Performance Evaluation and Comparisons". Application example of the proposed GFOD is given in "Application Example". Finally, "Conclusion" highlights the conclusions of the article.

## RELATED WORKS

This section reviews prior research on design methods of FFODs, VFODs, and GFODs, with a focus on the cutting edges, major findings of the recent techniques, and unresolved gaps.

Among the design methods of FFODs, discretization methods are very popular due to their simplicity and computational efficiency. These methods are broadly categorized into direct (*Visweswaran, Varshney & Gupta, 2011*; *El-Khazali, 2014*; *Rajasekhar & Krishna, 2018*; *Vasi et al., 2019*; *Rajasekhar & Krishna, 2020*; *Rajasekhar, 2022*) and indirect schemes (*Krishna, 2011*; *Maione, 2013*; *Krishna, 2019*; *Matusiak, Bakala & Wojciechowski, 2020*; *Ali et al., 2024*). VFOD has been realized using a composite structure of FIR subfilters whose coefficients are computed based on series expansion (*Tseng, 2008*). Although simple and do not require iterative procedures, such methods result in approximations that are accurate only over narrow band of frequencies and, in general, they do not allow the user to specify the band of interest.

Optimization-based methods have emerged as a powerful paradigm in digital filter design, offering unparalleled flexibility and precision compared to traditional analytical methods. By formulating filter synthesis as a mathematical optimization problem, these methods enable designers to systematically balance competing objectives, *e.g.*, magnitude error, phase linearity, and computational complexity, while adhering to application-specific constraints such as hardware resource budgets. The mechanism typically involves defining a cost function that evaluates deviations of a digital filter frequency response from a desired response, and utilizing an iterative procedure (*e.g.*, a gradient descent or a metaheuristic) to minimize this function.

Gradient-based methods have been used for the design of FFODs, VFODs, and GFODs. For example, *Johansson (2013)* proposed FFODs optimized in the minimax sense using iterative reweighted $L_1$-norm minimization. The iterative convex optimization algorithm was employed to design FFODs in *Chen, Wei & Meng (2024)*. VFOD was optimized using Park-McClellan minimax method in *Chan, Shyu & Yang (2010)*, where this method are used to compute the optimal coefficient of FIR subfilters of the composite structure. Weighted least squares method was employed to optimize the coefficients of the FIR subfilters of VFOD (*Tseng, 2004*, *2006*; *Shyu, Pei & Chan, 2009*; *Tseng & Lee, 2011a*, *2011b*, *2012a*, *2012b*) and GFOD (*Tseng & Lee, 2015*). However, these methods replace the original non-convex problem by its convex approximation, leading to local optimal solutions. In addition, suitable initial solutions are essential for them to converge.

Metaheuristic algorithms, inspired by biological or physical processes, further extend optimization framework to handle non-convex, multi-modal, or non-differentiable design spaces. They have gained traction for optimizing FFODs. FIR FFODs have been optimized using hybrid shuffled frog leaping algorithm (*Mohan & Rao, 2021*), lightning attachment procedure optimization (*Bansal & Gill, 2023*), and quantum-inspired evolutionary

algorithm (*Siddiqui, Mani & Singh, 2024*). IIR FFODs have been proposed using GA (*Das et al., 2011*), Nelder-Mead simplex algorithm (*Rana et al., 2016*), cuckoo search algorithm (*Barsainya, Rawat & Kumar, 2016*), particle swarm optimization (PSO) (*Mahata et al., 2016*), moth-flame optimization algorithm (*Mahata et al., 2019*), flower pollination algorithm (*Mahata et al., 2018*), multi-verse optimizer (*Ali et al., 2020*), ant colony optimization (*Nayak et al., 2019*), grey wolf optimizer-based cuckoo search algorithm (*Mouhou & Badri, 2022*), quantum-inspired evolutionary algorithm (*Siddiqui, Mani & Singh, 2024*), WOA (*Nayak & Kumar, 2024*), gradient based optimizer (*Moqbel, Ali & Xiao, 2024*), honey badger algorithm (*Rajasekhar, 2024*), and quasi-chaotic opposition-based mayfly algorithm (*Dey, Roy & Sarkar, 2025*). Nevertheless, existing studies focus almost exclusively on FFODs, neglecting the more complex and application-relevant challenge of designing VFODs and GFODs, where the differentiation order must adapt dynamically to real-time operational demands. This oversight leaves a significant gap in the development of adaptive systems for modern signal processing, where tunable fractional-order responses are essential for handling time-varying environments.

To bridge this research gap, we undertook the exploration of novel metaheuristics tailored for the design of VFOD and GFOD. Among the popular and recent metaheuristics, our attention has been directed toward WOA due its simplicity and powerful search mechanisms. However, like other metaheuristics, WOA faces challenges such as falling into the local optima, premature convergence, and low population diversity. As a consequence, many WOA variants have been proposed recently (*Nadimi-Shahraki et al., 2023*), which improved the WOA search strategies using Lévy flight, chaotic map, opposition-based learning, inertia weight strategy, or others. Chaotic map techniques, such as logistic, sine, and tent (*Nadimi-Shahraki et al., 2023*), have been observed to boost WOA by augmenting population diversity. Adaptive inertia weight strategies, *e.g.*, based on an exponential function (*Liu & Zhang, 2022*), a modified Versoria function (*Cao et al., 2023*), and a cosine function (*Yang & Guan, 2024*), have been employed to prevent it from getting stuck in a suboptimal solution and improve the convergence speed. Unlike previous variants, we integrate a piecewise linear chaotic mapping and an adaptive inertia weight based on a tangent hyperbolic function into original WOA framework.

Other implementation methods of variable fractional order operators have been proposed based on fast Fourier transform and fast convolution operations (*Matusiak, 2020*) and recurrent network architecture based on GRU-type cells (*Puchalski, 2022*). However, FFT must be truncated to finite lengths, which introduces Gibbs phenomena, degrading accuracy, particularly at low frequencies. Moreover, FFT-convolution assumes a static frequency response, making dynamic adjustments to the differentiation order or bandwidth impractical without recomputing the entire spectral profile, which is a computationally prohibitive task. On the other hand, while neural networks excel in adaptive tuning for time-varying systems, their trade-offs in interpretability, stability, and computational overhead render them less practical for many applications.

# THE DESIGN OF GFODS

In this section, the design method of GFODs is presented. First, the design problems are formulated as optimization problems, and the proposed composite structures for realizing GFODs are presented. Then, the optimization algorithm used to solve the formulated optimization problems is described.

## Problem formulation and realization structures

The main goal is to design a digital approximation $H(z)$ (*i.e.*, a GFOD) whose frequency response, $H(e^{j\omega})$, follows $H_d(\omega)$ as close as possible. As per (*Tseng & Lee, 2015*), the design problem of GFOD can be simplified using the relation between Eqs. (1) and (5),

$$H_d(\omega) = c_1 F_c(\omega) + c_2 F_c(-\omega), \tag{6}$$

where

$$
\begin{aligned}
c_1 &= \frac{\sin\left(\frac{\pi}{2}(p+\theta)\right)}{\sin(p\pi)}, \\
c_2 &= \frac{\sin\left(\frac{\pi}{2}(p-\theta)\right)}{\sin(p\pi)}.
\end{aligned}
\tag{7}
$$

Note that, for real-valued systems, the frequency response must satisfy conjugate symmetry, *i.e.*, $F_c(-\omega) = F_c^*(\omega)$, where $F_c^*(\omega)$ is the complex conjugate of $F_c(\omega)$. This symmetry arises from the requirement that the impulse response of a real filter is real-valued in the time domain. Consequently, designing a filter $F(z)$ whose frequency response, $F(e^{j\omega})$, matches $F_c(\omega)$ inherently constrains $F_c(-\omega)$ to be fitted by the frequency response $F(z^{-1})$, $F(e^{-j\omega})$. Thus, the transfer function of GFOD can be given by

$$H(z) = c_1 F(z) + c_2 F(z^{-1}), \tag{8}$$

where its frequency response is obtained by substituting $z \to e^{j\omega}$,

$$H(e^{j\omega}) = c_1 F(e^{j\omega}) + c_2 F(e^{-j\omega}). \tag{9}$$

Moreover, the design problem of GFOD is reduced to the approximation problem of $F_c(\omega)$ using $F(e^{j\omega})$. In what follows, the design problems of GFOD with fixed magnitude response and variable phase response and GFOD with variable magnitude and phase response are considered.

### GFOD with fixed p and variable θ

The proposed GFOD with fixed magnitude response and variable phase response, *i.e.*, fixed $p$ and variable $\theta$, is given by

$$H_1(z, \theta) = c_1 F(z) + c_2 F(z^{-1}). \tag{10}$$

The frequency response of $H_1(z, \theta)$ is given by

$$H_1(e^{j\omega}, \theta) = c_1 F(e^{j\omega}) + c_2 F(e^{-j\omega}). \tag{11}$$

The design problem of $H_1(z, \theta)$ is reduced to the design problem of $F(z)$ such that $F(e^{j\omega})$ well approximates $F_c(\omega)$ with fixed $p$. Since $p$ is fixed, then $F(z)$ is FFOD, which can be easily designed.

In *Tseng & Lee (2015)*, $F(z)$ is realized using an FIR structure whose coefficients are optimized using weighted least squares. In contrast, we realize $F(z)$ using an IIR transfer function that are computed to approximate $F_c(\omega)$ over a frequency band of interest $[\omega_1, \omega_2]$ as well as possible. The IIR transfer function used in this work is given by

$$F(z) = \frac{\sum_{n=0}^{L} b(n)z^{-n}}{1 + \sum_{n=1}^{L} a(n)z^{-n}}, \tag{12}$$

where the coefficients are computed to minimize the $L_p$-norm between $F_c(\omega)$ and the frequency response of $F(z)$ given by

$$F(e^{j\omega}) = \frac{\sum_{n=0}^{L} b(n)e^{-jn\omega}}{1 + \sum_{n=1}^{L} a(n)e^{-jn\omega}}. \tag{13}$$

Thus, the design procedure of GFOD with fixed $p$ and variable $\theta$ can be broken into the following steps. First, the coefficients of $F(z)$ are optimized to minimize the objective function,

$$J_1 = \int_{\omega_1}^{\omega_2} |F_c(\omega) - F(e^{j\omega})|^2 d\omega. \tag{14}$$

Second, the computed $F(z)$ and its conjugate $F(z^{-1})$ are substituted in Eq. (10), which yields

$$H_1(z, \theta) = c_1 \frac{\sum_{n=0}^{L} b(n)z^{-n}}{1 + \sum_{n=1}^{L} a(n)z^{-n}} + c_2 \frac{\sum_{n=0}^{L} b(n)z^{n}}{1 + \sum_{n=1}^{L} a(n)z^{n}}. \tag{15}$$

The proposed $H_1(z, \theta)$ can be realized using the composite structure given in Fig. 2. From this figure, the parameters $c_1$ and $c_2$ can be tuned to change the phase response of GFOD, without the need to recompute $F(z)$ and $F(z^{-1})$.

### GFOD with variable p and θ

The proposed GFOD with variable $p$ and variable $\theta$ is given by

$$H_2(z, p, \theta) = c_1 F(z, p) + c_2 F(z^{-1}, p). \tag{16}$$

The frequency response $H_2(z, p, \theta)$ is given by

$$H_2(e^{j\omega}, p, \theta) = c_1 F(e^{j\omega}, p) + c_2 F(e^{-j\omega}, p). \tag{17}$$

The design problem of $H_2(z, p, \theta)$ is reduced to the design problem of $F(z, p)$ such that using $F(e^{j\omega}, p)$ well approximates $F(\omega)$ with variable $p$. Since $p$ is variable, then $F(z, p)$ is VFOD, which can be realized using Farrow structure,

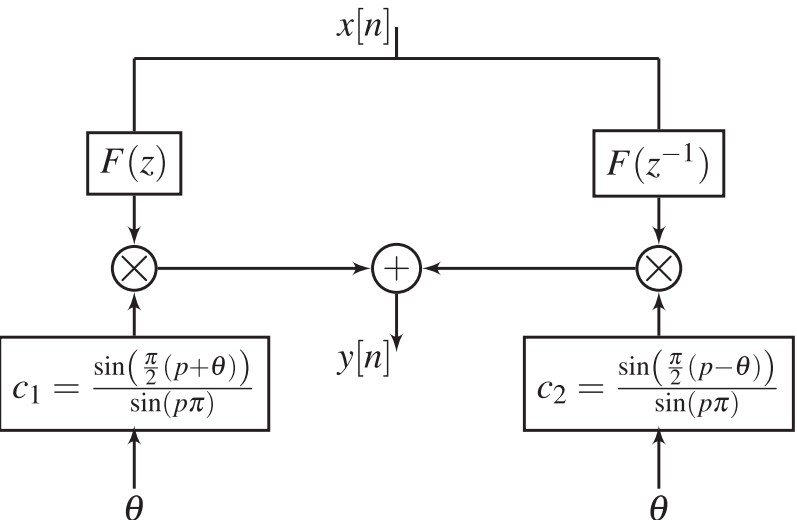

**Figure 2 Realization of GFOD with fixed $p$ and variable $\theta$, i.e., $H_1(z,\theta)$.**

$$F(z,p) = \sum_{k=0}^{M} A_k(z)p^k, \tag{18}$$

where $A_k(z)$, $k = 0, \cdots, M$, are subfilters.

In *Tseng & Lee (2015)*, $A_k(z)$ are realized using FIR structure whose coefficients are optimized using least squares method. In this work, we use IIR transfer functions to realize the subfilters $A_k(z)$, $k = 0, \cdots, M$. The utilized IIR transfer function is given by

$$A_k(z) = \frac{\sum_{n=0}^{L} b(n,k)z^{-n}}{1 + \sum_{n=1}^{L} a(n,k)z^{-n}}. \tag{19}$$

The proposed $F(z,p)$ is given by substituting in Eq. (19) in Eq. (18), which yields

$$F(z,p) = \sum_{k=0}^{M} p^k \frac{\sum_{n=0}^{L} b(n,k)z^{-n}}{1 + \sum_{n=1}^{L} a(n,k)z^{-n}}. \tag{20}$$

It follows that the coefficients $b(n,k)$ and $a(n,k)$ in Eq. (20) are computed to minimize the $L_p$-norm between $F_c(\omega)$ with variable $p$ and the frequency response of $F(z,p)$ given by

$$F(e^{j\omega},p) = \sum_{k=0}^{M} p^k \frac{\sum_{n=0}^{L} b(n,k)e^{-jn\omega}}{1 + \sum_{n=1}^{L} a(n,k)e^{-jn\omega}}. \tag{21}$$

Thus, the design procedure of GFOD with variable $p$ and $\theta$ can be broken into the following steps. First, the coefficients $b(n,k)$ and $a(n,k)$ of $F(z,p)$ are optimized to minimize the objective function,

$$J_2 = \int_{p_1}^{p_2} \int_{\omega_1}^{\omega_2} |F_c(\omega) - F(e^{j\omega},p)|^2 d\omega dp, \tag{22}$$

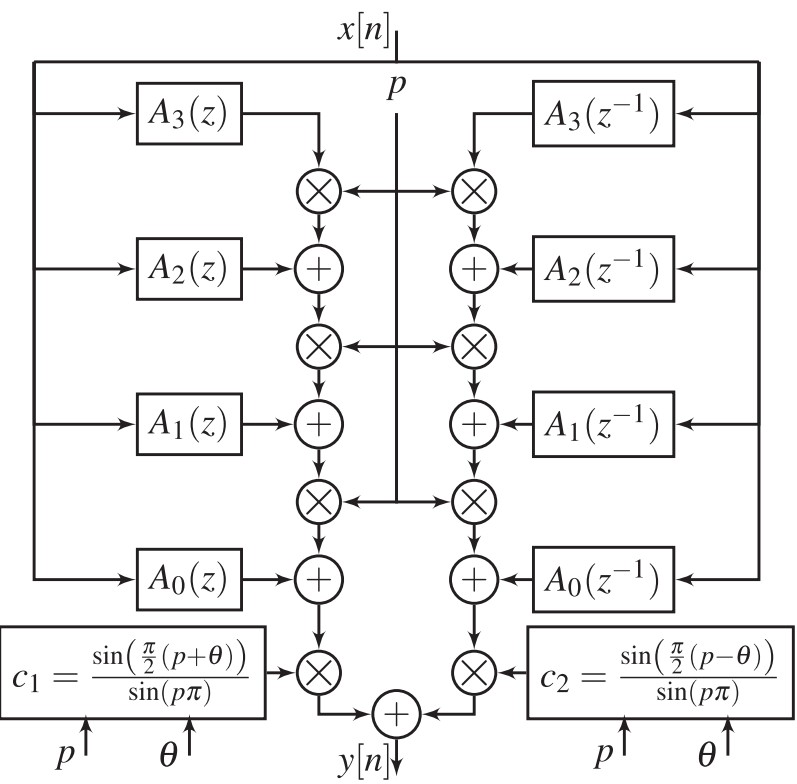

**Figure 3** Realization of GFOD with variable $p$ and variable $\theta$, i.e., $H_2(z, p, \theta)$.

where $[p_1, p_2]$ are the interested order range. Second, the computed $F(z, p)$ and its conjugate $F(z^{-1}, p)$ are substituted in Eq. (16), which yields

$$H_2(z, p, \theta) = c_1 \sum_{k=0}^{M} p^k \frac{\sum_{n=0}^{L} b(n, k)z^{-n}}{1 + \sum_{n=1}^{L} a(n, k)z^{-n}} + c_2 \sum_{k=0}^{M} p^k \frac{\sum_{n=0}^{L} b(n, k)z^n}{1 + \sum_{n=1}^{L} a(n, k)z^n}. \tag{23}$$

The realization of $H_2(z, p, \theta)$ is shown in Fig. 3. From this figure, the parameters $c_1$, $c_2$, and $p$ can be tuned to simultaneously change the magnitude and phase responses of GFOD, without the need to recompute the subfilters $A(z)$ and $A(z^{-1})$.

## Optimization algorithm

The proposed design method utilizes an improved version of WOA (IWOA) to solve the optimization problems formulated so far. The proposed IWOA is developed by integrating of PWLCM and AIWHT into original WOA framework to enhance address the drawbacks of WOA, i.e., low population diversity, slow convergence, and local optima entrapment.

WOA is a swarm-inspired metaheuristic that has recently been proposed by *Mirjalili & Lewis (2016)*. It is inspired by the hunting strategies of humpback whales, where it simulates encircling prey, searching for prey, and bubble-net foraging behavior of humpback whales. Given $N_x$ whales with positions $\mathbf{X} = \{\vec{X}_i, i = 1, 2, \cdots, N_x\}$ initialized randomly in the search space, the mathematical model of these strategies is summarized below.

### Encircling prey

The strategy of humpback whales to encircle the prey is mathematically modeled using the following equations:

$$\vec{D} = |\vec{C}\vec{X}^*(t) - \vec{X}(t)|, \tag{24}$$

$$\vec{X}(t+1) = \vec{X}^*(t) - \vec{A} \cdot \vec{D}, \tag{25}$$

where $t$ is the current iteration and $\vec{X}^*$ is the best position obtained so far. Since the global optimum is not known in *a priori*, WOA considers the best solution $\vec{X}^*$ as the prey position. $\vec{A}$ and $\vec{C}$ are given by

$$\begin{aligned} \vec{A} &= 2\vec{a} \cdot \vec{r}_1 - \vec{a}, \\ \vec{C} &= 2\vec{r}_2, \end{aligned} \tag{26}$$

where $\vec{r}_1$ and $\vec{r}_2$ are randomly chosen from $[0, 1]$, and

$$\vec{a} = 2 - \frac{2t}{\mathcal{T}}, \tag{27}$$

where $t$ is the current iteration number and $\mathcal{T}$ is the maximum number of iterations. It should be pointed out that $\vec{a}$ is linearly decreased from 2 to 0 over the course of iterations to emphasize exploration and exploitation for WOA. These two phases are formulated as follows.

### Exploration phase

The strategy of humpback whales to search for a prey is stimulated to constitute the exploration phase of WOA. The iterations with $|\vec{A}| > 1$ are devoted for exploration, where $\vec{A}$ is fluctuated over $[-a, a]$ as per Eq. (26). Here, one position from **X** is randomly chosen as a reference, $\vec{X}_{rand}$. Then, the remaining positions are updated with respect to $\vec{X}_{rand}$, rather than $\vec{X}^*$, using the following equations:

$$\vec{D} = |\vec{C} \cdot \vec{X}_{rand} - \vec{X}|, \tag{28}$$

$$\vec{X}(t+1) = \vec{X}_{rand} - \vec{A} \cdot \vec{D}. \tag{29}$$

Note that $|\vec{A}| > 1$ moves positions far away from $\vec{X}_{rand}$, which enhances the global search of WOA and emphasizes exploration.

### Exploitation phase

This phase is constructed by stimulating the bubble-net attack of humpback whales. In this attack, the whales use shrinking encircling mechanism along with a spiral-shaped movement to hunt the prey. To stimulate these two simultaneous strategies, WOA uses a parameter $pr$, that randomly chosen in the range $[0, 1]$ over the course of iterations, to decide either to update the positions of whales using shrinking circle (if $pr < 0.5$) or to update them using spiral model (if $pr \geq 0.5$). The mathematical model of this phase is done through the following equation:

$$\vec{X}(t+1) = \begin{cases} \vec{X}^*(t) - \vec{A} \cdot \vec{D}, & pr < 0.5 \\ \vec{D}'e^{bl} \cdot \cos(2\pi l) + \vec{X}^*(t), & pr \geq 0.5, \end{cases} \tag{30}$$

where $\vec{D} = |\vec{X}^*(t) - \vec{X}(t)|$ is the distance of the $\vec{X}$ to $\vec{X}^*$, $l$ is randomly chosen within $[1, 1]$, and $b$ is a constant that determines the spiral shape.

In summary, WOA starts with $n$ search agents that randomly positioned in the search space. As $a$ is linearly deceased from 2 to 0 over the course of iterations, it divides the iterations for exploration and exploitation. When $|\vec{A}| \geq 1$, WOA performs an exploration of the search space by moving the positions of the seach agents far away from a randomly chosen search agent. When $\vec{A} < 1$, WOA performs an exploitation around the best search agent found so far. This mechanism promotes a smooth transition between exploration and exploitation, which assists WOA to converge to a global optimum.

However, original WOA faces challenges such as falling into the local optima, premature convergence, and low population diversity. This article proposes an improve WOA by incorporating chaotic mapping and an adaptive inertia weight to enrich the population diversity, enhance the convergence speed, and boost the local optima escape.

### Piecewise linear chaotic map

Original WOA uses random initialization of the population, which may results in low diversity of the population. To increase the population diversity, researchers resorted to use chaotic maps such as logistic, sine, Chebyshev, circle, iterative, and skew tent map (*Nadimi-Shahraki et al., 2023*). In this article, we use piecewise linear chaotic map (PWLCM) to initialize the population. PWLCM is a segmented mapping function, where it utilizes an initial position as a segmentation indicator and then uses various formulas to calculate individuals at various positions. The compact formula for the PWLCM is given by

$$
x_{i+1} = \begin{cases} \frac{x_i}{d}, & 0 \leq x_i \leq d, \\ \frac{x_i - d}{0.5 - d}, & d < x_i \leq 0.5, \\ \frac{1 - x_i - d}{0.5 - d}, & 0.5 < x_i \leq 1-d, \\ \frac{1 - x_i}{d}, & 1-d < x_i \leq 1, \end{cases} \tag{31}
$$

where $d$ is a control parameter used to determine the segmentation range of the four-segment formula. The value $d = 0.3$ was used in all experiments in this study. Based on Eq. (31), the elements $x_{ij}$, $i = 1, \cdots, N_x$ and $j = 1, \cdots, Q$, where $Q$ is the search space dimensions, are generated. The initial population is then computed by

$$
X_{ij} = (Ub - Lb)x_{ij} + Lb, \tag{32}
$$

where $Ub$ and $Lb$ are the upper and lower boundaries, respectively.

### Adaptive inertia weight based on the hyperbolic tangent

In original WOA, the search agents get closer to those with better fitness values as the optimization progresses. However, this increases the chance of premature convergence and local optima entrapment. This article attempts to address this problem by adjusting the population positions using an adaptive inertia weight mechanism. It is important to know that higher weight is preferred in the early stage of optimization process to boost the global exploration, while lower weight is favorable in the late stage to fine-tune the local search region (*Liu et al., 2022*). We use the hyperbolic tangent function, $\tanh(q)$, where

---

**Algorithm 1** **The pseudo code of IWOA.**

Initialize the whales population $X_i(i = 1, 2, \cdots, N_x)$ using Eq. (32)
Calculate the fitness of each search agent
$\vec{X}^*$ = the best search agent
**while** $t < \mathcal{T}$ **do**
    **for** each search agent **do**
        Update $\vec{a}$, $\vec{A}$, $\vec{C}$, $l$, and $pr$
        **if** $pr < 0.5$ **then**
            **if** $|\vec{A}| \geq 1$ **then**
                Select a random search agent ($\vec{X}_{rand}$)
                Update $\vec{X}(t + 1)$ by Eq. (29)
            **else if** $|A| < 1$ **then**
                Update $\vec{X}(t + 1)$ by Eq. (34)
            **end if**
        **else if** $pr \geq 0.5$ **then**
            Update $\vec{X}(t + 1)$ by Eq. (34)
        **end if**
    **end for**
    Calculate the fitness of each search agent
    Update $\vec{X}^*$ if there is a better solution
    $t = t + 1$
**end while**
**return** $X^*$

---

$q \in [-5, 5]$, to control the inertia weight coefficient for nonlinearly adaptive adjustment (*Liu et al., 2022*), *i.e.*,

$$w(t) = \frac{w_{max} + w_{min}}{2} + \frac{w_{max} - w_{min}}{2} \times \tanh\left(-5 + 10\frac{\mathcal{T} - t}{\mathcal{T}}\right), \tag{33}$$

where $w_{max}$ and $w_{min}$ are the maximum and minimum inertia weights, where we set to $w_{max} = 0.9$ and $w_{min} = 0.4$. Then, incorporating the weight into Eq. (30), we get

$$\vec{X}(t + 1) = \begin{cases} w(t)\vec{X}^*(t) - \vec{A} \cdot \vec{D}, pr < 0.5 \\ \vec{D'}e^{bl} \cdot \cos(2\pi l) + w(t)\vec{X}^*(t), pr \geq 0.5, \end{cases} \tag{34}$$

The pseudo code of IWOA is given in Algorithm 1.

# PERFORMANCE EVALUATION AND COMPARISONS

In this section, simulation experiments for two design examples are conducted to evaluate the performance of the proposed design method for GFOD with respect to well-known techniques: real-coded GA (RCGA), PSO, and original WOA. The first example concerns the design of GFOD with fixed $p$ and variable $\theta$, and the second example concerns the design of GFOD with variable $p$ and $\theta$. In both examples, the frequency band of interest is

**Table 1 Control parameters for the metaheuristics being compared.**

| Parameter | RCGA | PSO | WOA | IWOA |
|---|---|---|---|---|
| Size of population ($N_x$) | 50 | 50 | 50 | 50 |
| Max. number of iterations ($\mathscr{T}$) | 500 | 500 | 500 | 500 |
| Crossover operator | Two-point | – | – | – |
| Mutation rate | 0.1 | – | – | – |
| Mechanism of selection | Roulette wheel mechanism | – | – | – |
| Cognitive and social constants | – | 2.0, 2.0 | – | – |
| Velocity limits | – | 0.05-1 | – | – |
| Inertia weight limits | – | 0.4-0.9 | – | 0.4-0.9 |

chosen as $[\omega_1 = 0.05\pi, \omega_2 = 0.95\pi]$. The performance metrics are the frequency response error *versus* $\omega \in [0.05\pi, 0.95\pi]$ and the normalized root mean squares error, which are given below for each design example. To have a robust comparisons, the results of each design example are obtained based on 100 trial independent runs for each algorithm. The control parameters for RCGA, PSO, WOA, and IWOA are listed in Table 1. The simulations have been carried out in MATLAB 9.0 (The MathWorks, Natick, MA, USA) operating on Intel(R) Core(TM), i7-4510U CPU @ 2.6 GHz and 8 GB RAM. Further, the proposed GFODs are compared with state-of-the-art ones in terms of frequency response error, normalized root mean squares error, and computational complexity.

## GFOD with fixed $p$ and variable $\theta$

In this example, the design of GFOD with fixed $p$ and variable $\theta$, *i.e.*, $H_1(e^{j\omega}, \theta)$, is considered. The performance of $H_1(e^{j\omega}, \theta)$ is evaluated based on frequency response error,

$$Err_1(\theta) = \left( \int_{\omega_1}^{\omega_2} |H_1(e^{j\omega}, \theta) - H_d(\omega)|^2 d\omega \right)^{\frac{1}{2}}, \tag{35}$$

and normalized root mean squares (NRMS) error,

$$E_1 = \left( \frac{\int_{\theta_1}^{\theta_2} \int_{\omega_1}^{\omega_2} |H_1(e^{j\omega}, \theta) - H_d(\omega)|^2 d\omega d\theta}{\int_{\theta_1}^{\theta_2} \int_{\omega_1}^{\omega_2} |H_d(\omega)|^2 d\omega d\theta} \right)^{\frac{1}{2}}. \tag{36}$$

The fractional order is set as $p = 0.5$, and the order of the IIR subfilters is set as $L = 8$. To ensure that the phase angle of $H_1(e^{j\omega}, \theta)$ can be any value between $-\pi$ and $\pi$, the range of $\theta$ is set as $[\theta_1 = -2, \theta_2 = 2]$. The magnitude and normalized phase responses of the best RCGA-, PSO-, WOA-, and IWOA-based designs of $H_1(e^{j\omega}, \theta)$ are depicted in Fig. 4. Apparently, all designs of $H_1(e^{j\omega}, \theta)$ accurately follow the ideal FOD over most of the frequency band of interest. The respective error $Err_1(\theta)$ are shown in Fig. 5, which is observed to consistently has maximum at $\theta = \pm 1$ and minimum at $\theta = 0$ and $\theta = \pm 2$. This figure demonstrates that the IWOA-based design significantly outperforms the

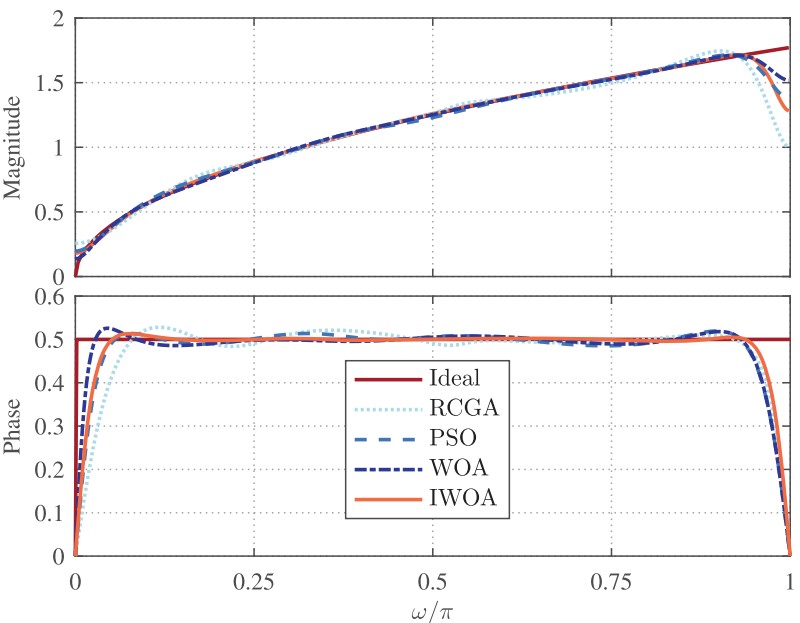

**Figure 4 Magnitude and normalized phase responses of $H_1(e^{j\omega}, \theta)$ based on RCGA, PSO, WOA, and IWOA.**

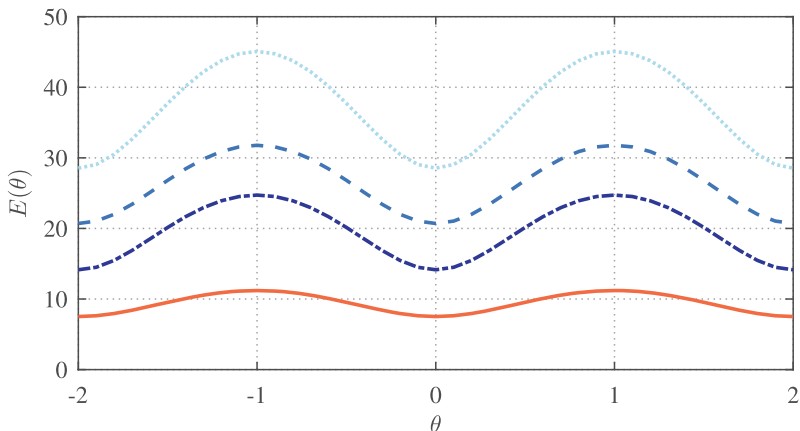

**Figure 5 $Err_1(\theta)$ for $H_1(e^{j\omega}, \theta)$ based on RCGA, PSO, WOA, and IWOA.**

RCGA-, PSO-, and WOA-based designs in terms of magnitude response. In addition, the absolute normalized phase error of the RCGA-, PSO-, WOA-, and IWOA-based designs are given in Fig. 6. It can be seen from these figures that the IWOA-based design achieves more accurate phase response than the others. Moreover, the NRMS error is computed for all designs and shown to be 0.9065%, 0.5211%, 0.5163%, and 0.1720% for the RCGA-, PSO-, and WOA-, and IWOA-based designs, respectively.

The proposed IWOA-based GFOD can be compared to the FIR-based one of *Tseng & Lee (2015)* obtained using the least squares method. To see the performance of the *Tseng &*

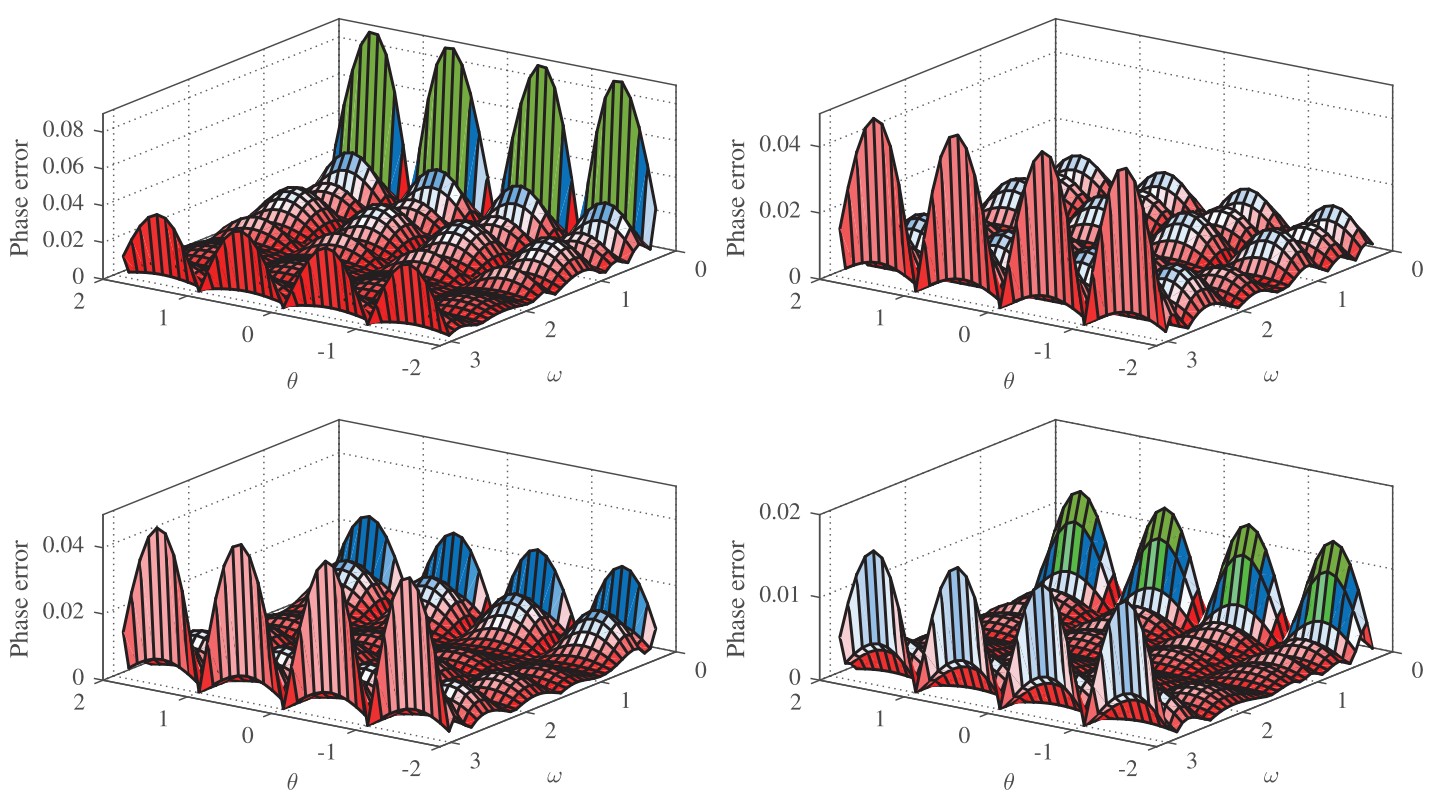

**Figure 6** Phase error for $H_1(e^{j\omega}, \theta)$ based on RCGA (top-left), PSO (top-right), WOA (bottom-left), and IWOA (bottom-right).

*Lee*'s *(2015)* design, the reader is referred to the results shown in Fig. 2 and Fig. 3 in *Tseng & Lee (2015)*; the figures are not reproduced here since the coefficients are not provided in *Tseng & Lee (2015)*. It can be seen that the proposed GFOD is slightly worse than the one due to *Tseng & Lee (2015)* in terms of magnitude and phase responses. This can also be demonstrated by the NRMS error of 0.1352% of *Tseng & Lee*'s *(2015)* GFOD and that of our GFOD, 0.1720%. However, our design is much more efficient than the one by *Tseng & Lee (2015)*. It can be observed from Eq. (12) that for $L = 8$, then $F(z)$ requires 16 multipliers, 16 adders, and 16 delay elements. On the other hand, in *Tseng & Lee (2015)*, the FIR subfilter, with $L = 30$, requires 30 multipliers, 30 adders, and 29 delay elements. Thus, our GFOD saves about 50% of implementation complexity compared to the one due to *Tseng & Lee (2015)*.

Moreover, we conduct a comparison of the overall performance of IWOA with those of RCGA, PSO, WOA, WOA with PWLCM, and WOA with AIWHT through an average-, best- and worst-case analysis over the entire independent runs, and summarize the results in Table 2. As it is clear from these results, IWOA outperforms the other algorithms in average, and hence consistently superior over them in designing GFODs with fixed $p$ and variable $\theta$. Also, IWOA achieves more stable performance than the others since it has smaller standard deviation values.

**Table 2 Performance metrics of the RCGA, PSO, WOA, WOA with PWLCM, WOA with AIWHT, and IWOA-based GFODs of fixed $p$ and variable $\theta$.**

|  | $L=4$ | | | | $L=6$ | | | | $L=8$ | | | |
|---|---|---|---|---|---|---|---|---|---|---|---|---|
|  | min. | max. | ave. | std. | min. | max. | ave. | std. | min. | max. | ave. | std. |
| RCGA | 1.6282 | 8.5199 | 5.1195 | 2.0445 | 1.4529 | 7.2696 | 4.3926 | 1.5906 | 0.9065 | 4.5234 | 2.4977 | 1.0065 |
| PSO | 0.8410 | 6.0384 | 3.3191 | 1.5015 | 0.6974 | 5.5210 | 3.2980 | 1.4442 | 0.5211 | 4.8381 | 2.7146 | 1.2521 |
| WOA | 0.8997 | 5.7053 | 3.1297 | 1.3998 | 0.6686 | 3.6956 | 2.3283 | 0.8426 | 0.5163 | 2.9345 | 1.6969 | 0.7136 |
| WOA-PWLCM | 0.5654 | 2.9842 | 1.7029 | 0.6712 | 0.3863 | 1.6280 | 1.0521 | 0.3206 | 0.3225 | 1.2485 | 0.7636 | 0.2748 |
| WOA-AIWHT | 0.4090 | 2.0113 | 1.2633 | 0.4600 | 0.3824 | 1.7292 | 1.0386 | 0.3858 | 0.2844 | 1.3557 | 0.8516 | 0.3020 |
| IWOA | 0.3316 | 1.5041 | 0.9756 | 0.3368 | 0.2417 | 1.1189 | 0.7024 | 0.2752 | 0.1720 | 0.8045 | 0.4988 | 0.1845 |

Further, we examine how adjusting the two global control parameters, population size ($N_x$) and maximum iteration count ($\mathcal{T}$), influences the performance of RGA, PSO, WOA, and IWOA algorithms in optimizing the design of GFODs. To this end, we re-conduct the experiments for $N_x = 50$ and $\mathcal{T} = 2{,}000$ and for $N_x = 100$ and $\mathcal{T} = 2{,}000$. On the basis of 100 independent runs for each design case, a comparison of the overall performance of RCGA, PSO, WOA, and IWOA is carried out through an average-, best- and worst-case analysis, and the results are summarized in Table 3. Similar to the case of $N_x = 50$ and $\mathcal{T} = 500$, IWOA performs better than the other alogirthms for each combination of $N_x$ and $\mathcal{T}$.

Interestingly, it is noted from Table 3 that increasing the maximum iteration limit ($\mathcal{T}$) to 2,000 does not substantially enhance the performance of the designed GFODs compared to those optimized with $\mathcal{T} = 500$. This suggests that exceeding 500 iterations leads to unnecessary computational overhead without meaningful gains in accuracy. A parallel observation applies to evaluations with a population size ($N_x$) of 100, which similarly fails to outperform results achieved with $N_x = 50$. Consequently, the optimal parameter configuration for the GFOD design problem is identified as $N_x = 50$ combined with $\mathcal{T} = 500$, which balances efficiency and effectiveness.

### GFOD with variable $p$ and $\theta$

In this example, the design of GFOD with variable $p$ and variable $\theta$, *i.e.*, $H_2(e^{j\omega}, p, \theta)$, is considered. The performance of $H_2(e^{j\omega}, p, \theta)$ is evaluated based on frequency response error,

$$Err_2(p, \theta) = \left( \int_{\omega_1}^{\omega_2} |H_2(e^{j\omega}, p, \theta) - H_d(\omega)|^2 d\omega \right)^{\frac{1}{2}} \tag{37}$$

and NRMS error,

$$E_2 = \left( \frac{\int_{p_1}^{p_2} \int_{\theta_1}^{\theta_2} \int_{\omega_1}^{\omega_2} |H_2(e^{j\omega}, \theta) - H_d(\omega)|^2 d\omega d\theta dp}{\int_{p_1}^{p_2} \int_{\theta_1}^{\theta_2} \int_{\omega_1}^{\omega_2} |H_d(\omega)|^2 d\omega d\theta dp} \right)^{\frac{1}{2}} \tag{38}$$

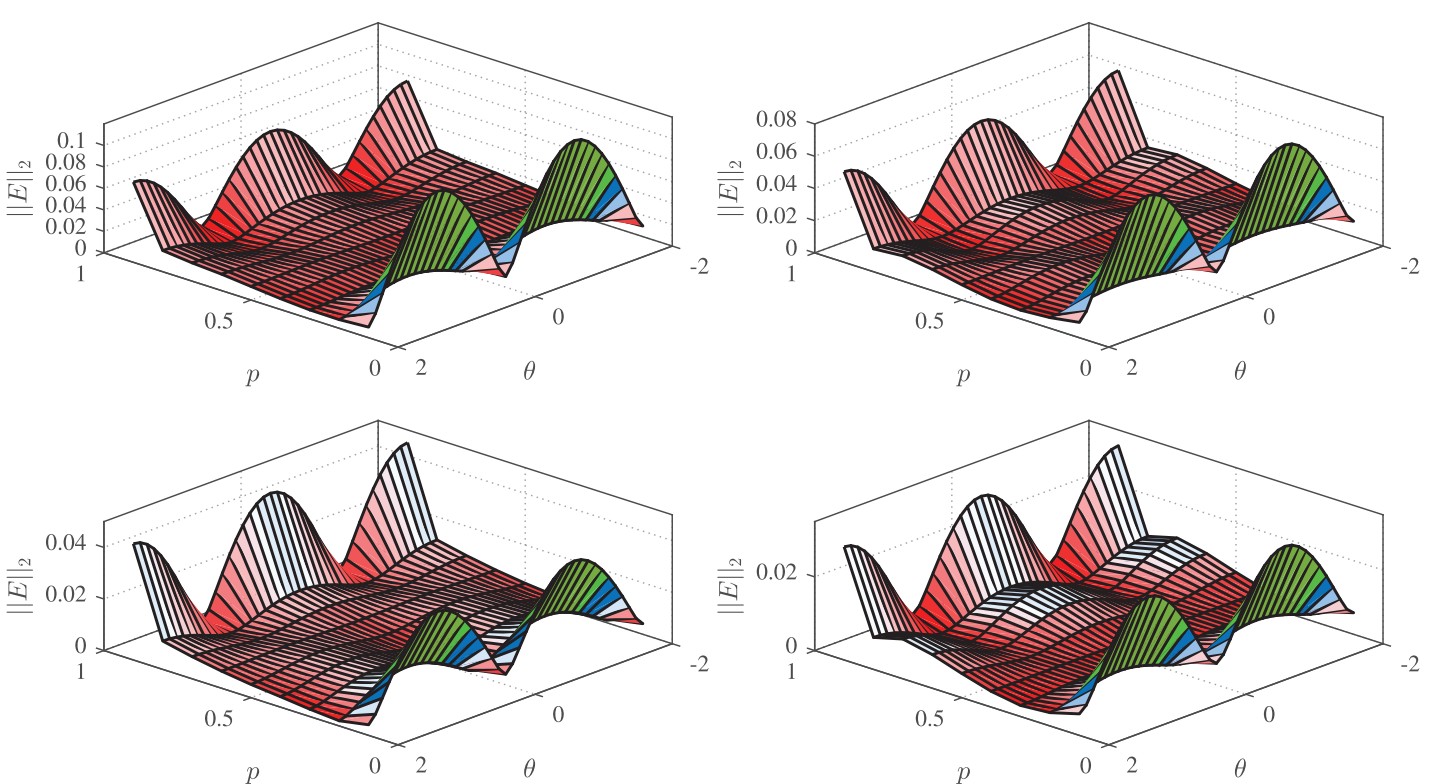

**Figure 7** $Err_2(p, \theta)$ for $H_2(e^{j\omega}, p, \theta)$ **based on RCGA (top-left), PSO (top-right), WOA (bottom-left), and IWOA (bottom-right).**

The range of fractional order is set as $[p_1 = 0, p_2 = 1]$, and the range of $\theta$ is set as $[\theta_1 = -2, \theta_2 = 2]$. The order of the IIR subfilters is set as $L = 6$, and the number of subfilters is set as $M = 5$. Note that our choice for $M$ aligns with the value adopted in *Tseng & Lee (2015)*, ensuring a consistent basis for comparative analysis and facilitating a fair evaluation of our design's performance against that of *Tseng & Lee*'s *(2015)* GFOD. Meanwhile, our selection of $L = 6$ is motivated by experimental evaluations demonstrating that, as we shall see later, this order optimally balances computational complexity and solution accuracy. The error $Err_2(p, \theta)$ for the RCGA-, PSO-, WOA-, and IWOA-based designs are given in Fig. 7. It can be seen from these figures that the IWOA-based design achieves more accurate magnitude response than the others. The NRMS error is computed for all designs and shown to be 0.3313%, 0.2529%, 0.2102, and 0.1683% for the RCGA-, PSO-, and WOA-, IWOA-based designs, respectively.

The proposed IWOA-based GFOD can be compared to the FIR-based one proposed in *Tseng & Lee (2015)* obtained using the least squares method. To see the performance of the *Tseng & Lee*'s *(2015)* design, the reader is referred to the results shown in Fig. 6 in *Tseng & Lee (2015)*. It can be seen that the proposed GFOD is competitive to the one due to *Tseng & Lee (2015)* in terms of magnitude and phase responses. The NRMS error for *Tseng & Lee*'s *(2015)* GFOD is 0.1182%, while that for our GFOD is 0.1683%. However, our design is much more efficient than that of *Tseng & Lee (2015)*. From Eq. (19), each IIR subfilter in

**Table 3 Performance metrics of the RCGA, PSO, WOA, and IWOA-based GFODs of fixed $p$ and variable $\theta$ for different combinations of population size and maximum number of iterations.**

| $N_x$ | $\mathscr{T}$ | | $L = 4$ | | | | $L = 6$ | | | | $L = 8$ | | | |
|---|---|---|---|---|---|---|---|---|---|---|---|---|---|---|
| | | | min. | max. | ave. | std. | min. | max. | ave. | std. | min. | max. | ave. | std. |
| 50 | 500 | RCGA | 1.6282 | 8.5199 | 5.1195 | 2.0445 | 1.4529 | 7.2696 | 4.3926 | 1.5906 | 0.9065 | 4.5234 | 2.4977 | 1.0065 |
| | | PSO | 0.8410 | 6.0384 | 3.3191 | 1.5015 | 0.6974 | 5.5210 | 3.2980 | 1.4442 | 0.5211 | 4.8381 | 2.7146 | 1.2521 |
| | | WOA | 0.8997 | 5.7053 | 3.1297 | 1.3998 | 0.6686 | 3.6956 | 2.3283 | 0.8426 | 0.5163 | 2.9345 | 1.6969 | 0.7136 |
| | | IWOA | 0.3316 | 1.5041 | 0.9756 | 0.3368 | 0.2417 | 1.1189 | 0.7024 | 0.2752 | 0.1720 | 0.8045 | 0.4988 | 0.1845 |
| 50 | 2,000 | RCGA | 1.7952 | 8.6646 | 5.1898 | 2.0220 | 1.3844 | 7.2467 | 4.4258 | 1.5952 | 0.8632 | 4.6538 | 2.5578 | 1.0164 |
| | | PSO | 0.6850 | 6.1514 | 3.3977 | 1.5270 | 0.6741 | 5.5360 | 3.3394 | 1.4375 | 0.4442 | 4.8987 | 2.7597 | 1.2592 |
| | | WOA | 0.8287 | 5.8186 | 3.1834 | 1.4118 | 0.6746 | 3.7480 | 2.3739 | 0.8505 | 0.5280 | 3.0133 | 1.7554 | 0.7151 |
| | | IWOA | 0.3201 | 1.8349 | 1.0445 | 0.3780 | 0.2437 | 1.2463 | 0.7479 | 0.2874 | 0.1004 | 0.9448 | 0.5679 | 0.2028 |
| 100 | 2,000 | RCGA | 1.6978 | 8.5612 | 5.1428 | 2.0464 | 1.4091 | 7.2430 | 4.3698 | 1.5913 | 0.9294 | 4.5322 | 2.5081 | 0.9984 |
| | | PSO | 0.8348 | 6.0395 | 3.3406 | 1.4967 | 0.6991 | 5.5392 | 3.2781 | 1.4416 | 0.5647 | 4.8548 | 2.7229 | 1.2522 |
| | | WOA | 0.8782 | 5.7055 | 3.1497 | 1.3985 | 0.6238 | 3.7033 | 2.3104 | 0.8442 | 0.5342 | 2.9194 | 1.7062 | 0.7136 |
| | | IWOA | 0.3483 | 1.5378 | 0.9982 | 0.3345 | 0.2139 | 1.1176 | 0.6851 | 0.2812 | 0.1444 | 0.8453 | 0.5070 | 0.1913 |

our GFOD, with $L = 6$, requires 12 multipliers, 12 adders, and 10 delay elements. On the other hand, each FIR subfilter in *Tseng & Lee*'s *(2015)* design, with $L = 30$, requires 30 multipliers, 30 adder, 29 delay elements. Thus, our GFOD saves more than 50% of implementation complexity compared to the one due to *Tseng & Lee (2015)*.

Similar to fixed $p$ case, we conduct a comparison of the overall performance of IWOA with those of RCGA, PSO, WOA, WOA with PWLCM, and WOA with AIWHT through an average-, best- and worst-case analysis over the entire independent runs, and summarize the results in Table 4. As it is clear from these results, IWOA outperforms the other algorithms in average, and hence consistently superior over them in designing GFODs with variable $p$ and $\theta$. Also, IWOA achieves more stable performance than the others since it has smaller standard deviation values.

Further, we conduct a Wilcoxon rank-sum statistical test over the entire independent runs in terms NRMS values to compare the performance of IWOA and those of the benchmarks. We use a pair-wise combinations of $n_1$ and $n_2$ outcomes, where $n_1$ and $n_2$ NRMS values are randomly chosen from those of each benchmark algorithm and IWOA respectively. We then compare the respective sum of ranks (*i.e.*, $W_1$ and $W_2$) with the critical values $W_{th}$ given in (*Montgomery & Runger, 2011*) to determine at what level of significance, $a$, the null hypothesis is reject or it is rather accepted. The confidence level of the difference between the outcomes is $CL = (1-a) \times 100\%$. We summarize the results of the conducted statistical test in Table 5. The results demonstrate the consistent superiority of IWOA over the benchmarks in the design of GFODs, where the null hypothesis is rejected with high levels of confidence in most design cases.

Finally, it is worth mentioning that the proposed IWOA, integrating a PWLCM and a AIWHT mechanism, retains the asymptotic computational complexity of the conventional WOA, *i.e.*, $O(\mathscr{T} \times N_x \times Q)$, where $\mathscr{T}$, $N_x$, and $Q$ denote iterations, population size, and

**Table 4 Performance metrics of the RCGA, PSO, WOA, WOA with PWLCM, WOA with AIWHT, and IWOA-based GFODs of variable $p$ and $\theta$.**

| | | L = 4 | | | | L = 6 | | | | L = 8 | | | |
|---|---|---|---|---|---|---|---|---|---|---|---|---|---|
| | **M** | min. | max. | ave. | std. | min. | max. | ave. | std. | min. | max. | ave. | std. |
| RCGA | 3 | 0.9517 | 3.2220 | 2.0663 | 0.6324 | 0.8027 | 3.8062 | 2.2256 | 0.8370 | 0.7683 | 4.0332 | 2.4880 | 0.9782 |
| | 5 | 0.4200 | 1.4247 | 0.9580 | 0.2927 | 0.3313 | 1.8255 | 1.1229 | 0.4673 | 0.3485 | 2.5801 | 1.4916 | 0.6740 |
| PSO | 3 | 0.6834 | 2.9193 | 1.8994 | 0.6813 | 0.5630 | 2.9717 | 1.7770 | 0.6475 | 0.4491 | 3.1788 | 1.8330 | 0.8030 |
| | 5 | 0.3783 | 1.1559 | 0.7284 | 0.2265 | 0.2529 | 1.1864 | 0.7543 | 0.2680 | 0.2357 | 1.1685 | 0.7121 | 0.2845 |
| WOA | 3 | 0.6206 | 2.4897 | 1.4976 | 0.5410 | 0.5156 | 2.4771 | 1.5287 | 0.6039 | 0.4502 | 2.7723 | 1.6442 | 0.6650 |
| | 5 | 0.3699 | 1.1944 | 0.7828 | 0.2445 | 0.2102 | 1.1994 | 0.7105 | 0.2902 | 0.2241 | 1.1954 | 0.7445 | 0.2855 |
| WOA-PWLCM | 3 | 0.5009 | 1.9415 | 1.2360 | 0.3996 | 0.4439 | 1.9582 | 1.1180 | 0.4496 | 0.4184 | 2.1276 | 1.3196 | 0.5082 |
| | 5 | 0.3222 | 0.8878 | 0.6169 | 0.1636 | 0.1972 | 0.8737 | 0.5387 | 0.2103 | 0.1884 | 0.8800 | 0.5296 | 0.2023 |
| WOA-AIWHT | 3 | 0.4886 | 2.0553 | 1.2384 | 0.4543 | 0.4667 | 2.0100 | 1.1397 | 0.4289 | 0.4015 | 2.2509 | 1.3880 | 0.5142 |
| | 5 | 0.2951 | 0.8198 | 0.5549 | 0.1577 | 0.1936 | 0.8133 | 0.5002 | 0.1775 | 0.1909 | 0.8200 | 0.5194 | 0.1899 |
| IWOA | 3 | 0.3922 | 0.9917 | 0.6972 | 0.1585 | 0.2990 | 0.9989 | 0.6566 | 0.1961 | 0.3193 | 0.9876 | 0.6620 | 0.1839 |
| | 5 | 0.1795 | 0.5592 | 0.3811 | 0.1078 | 0.1683 | 0.5345 | 0.3600 | 0.1082 | 0.1604 | 0.5614 | 0.3709 | 0.1156 |

**Table 5 Wilcoxon rank-sum test over the entire independent runs.**

| | | | Wilcoxon rank-sum test | | | | | |
|---|---|---|---|---|---|---|---|---|
| | | | $n_1 = 7$ and $n_2 = 11$ (For $a = 0.05$ $W_{th} = 44$) (For $a = 0.01$ $W_{th} = 38$) | | | $n_1 = 10$ and $n_2 = 12$ (For $a = 0.05$ $W_{th} = 85$) (For $a = 0.01$ $W_{th} = 76$) | | |
| **M** | **L** | **Benchmark algorithm** | $W_1$ | $W_2$ | **Accept/reject (CL)** | $W_1$ | $W_2$ | **Accept/reject (CL)** |
| NA | 4 | RCGA | 28 | 143 | Reject (99%) | 55 | 198 | Reject (99%) |
| | | PSO | 40 | 131 | Reject (95%) | 58 | 195 | Reject (99%) |
| | | WOA | 31 | 140 | Reject (99%) | 58 | 195 | Reject (99%) |
| | 6 | RCGA | 28 | 143 | Reject (99%) | 55 | 198 | Reject (99%) |
| | | PSO | 34 | 137 | Reject (99%) | 61 | 192 | Reject (99%) |
| | | WOA | 28 | 143 | Reject (99%) | 55 | 198 | Reject (99%) |
| | 8 | RCGA | 28 | 143 | Reject (99%) | 55 | 198 | Reject (99%) |
| | | PSO | 28 | 143 | Reject (99%) | 55 | 198 | Reject (99%) |
| | | WOA | 28 | 143 | Reject (99%) | 56 | 197 | Reject (99%) |
| 3 | 4 | RCGA | 28 | 143 | Reject (99%) | 55 | 198 | Reject (99%) |
| | | PSO | 28 | 143 | Reject (99%) | 55 | 198 | Reject (99%) |
| | | WOA | 28 | 143 | Reject (99%) | 55 | 198 | Reject (99%) |
| | 6 | RCGA | 29 | 142 | Reject (99%) | 56 | 197 | Reject (99%) |
| | | PSO | 33 | 138 | Reject (99%) | 56 | 197 | Reject (99%) |
| | | WOA | 37 | 134 | Reject (99%) | 62 | 191 | Reject (99%) |
| | 8 | RCGA | 32 | 139 | Reject (99% | 56 | 197 | Reject (99%) |
| | | PSO | 47 | 124 | Accept | 84 | 169 | Reject (95%) |
| | | WOA | 66 | 105 | Accept | 101 | 152 | Accept |

| | | | Wilcoxon rank-sum test | | | | | |
| | | | $n_1 = 7$ **and** $n_2 = 11$ (**For** $a = 0.05$ $W_{th} = 44$) (**For** $a = 0.01$ $W_{th} = 38$) | | | $n_1 = 10$ **and** $n_2 = 12$ (**For** $a = 0.05$ $W_{th} = 85$) (**For** $a = 0.01$ $W_{th} = 76$) | | |
| $M$ | $L$ | Benchmark algorithm | $W_1$ | $W_2$ | Accept/reject (CL) | $W_1$ | $W_2$ | Accept/reject (CL) |
| --- | --- | --- | --- | --- | --- | --- | --- | --- |
| 5 | 4 | RCGA | 38 | 133 | Reject (99%) | 65 | 188 | Reject (99%) |
| | | PSO | 39 | 132 | Reject (95%) | 69 | 184 | Reject (99%) |
| | | WOA | 40 | 131 | Reject (95%) | 68 | 185 | Reject (99%) |
| | 6 | RCGA | 39 | 132 | Reject (95%) | 68 | 185 | Reject (99%) |
| | | PSO | 51 | 120 | Accept | 91 | 162 | Accept |
| | | WOA | 33 | 138 | Reject (99%) | 72 | 181 | Reject (99%) |
| | 8 | RCGA | 41 | 130 | Reject (95%) | 59 | 194 | Reject (99%) |
| | | PSO | 43 | 128 | Reject (95%) | 64 | 189 | Reject (99%) |
| | | WOA | 47 | 124 | Accept | 77 | 176 | Reject (95%) |

problem dimensionality, respectively. While PWLCM introduces a one-time overhead during initialization by iterating chaotic sequences for enhanced diversity, its complexity remains asymptotically linear $O(N_x \times Q)$, as the added operations involve only a constant factor. Similarly, AIWHT mechanism incurs a per-iteration cost of $O(N_x)$ for dynamically adjusting inertia weights *via* hyperbolic tangent function, but this term is dominated by the $O(N_x \times Q)$ position updates. Consequently, the dominant complexity term $O(\mathscr{T} \times N_x \times Q)$ remains unchanged. Practically, the modifications introduce minor runtime trade-offs, namely, PWLCM marginally slows initialization, and AIWHT adds lightweight scalar computations per iteration. However, these enhancements often reduce the effective iterations required for convergence by improving exploration-exploitation balance, thereby offsetting the overheads and yielding superior solution quality. Thus, while asymptotic equivalence persists, IWOA achieves better performance with negligible computational burden, aligning with trends in adaptive metaheuristic design.

## APPLICATION EXAMPLE

In this section, the performance of the proposed GFOD is experimentally verified in edge detection of a signal. A square pulse signal is used as an input for the proposed GFOD in Example 2, and the output $y(n)$,

$$y(n) = ifft(fft(x(n)). * H_2(e^{j\omega}, p, \theta)), \tag{39}$$

for various $p$ and $\theta$ is plotted. As noted in *Tseng & Lee (2015)*, the width and type of the detected edge can be controlled using $p$ and $\theta$ respectively. To confirm this in our study, the case of fixed $\theta$ and variable $p$ and the case of fixed $p$ and variable $\theta$ are considered. That is, the output signals for $\theta = 1$ and $p = 0.1, 0.5$, and $0.8$ are given in the top panel of Fig. 8, and the output signals for $p = 0.5$ and $\theta = 0.2, 1$, and $1.8$ are given in the bottom panel of Fig. 8. The outputs signals of ideal GFOD for all cases are also given in the respected sub

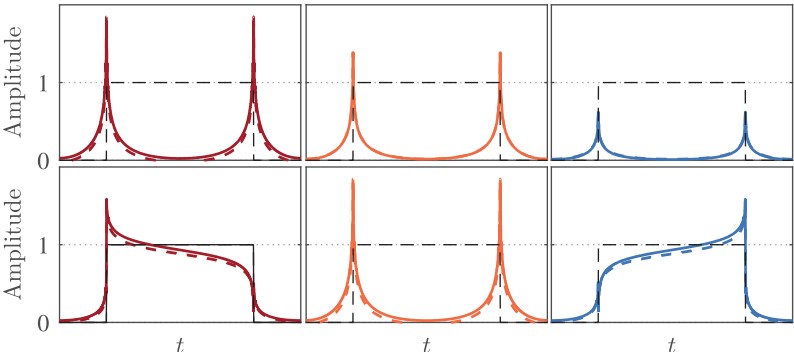

**Figure 8 Signal edge detection using the proposed GFOD in Example 2 with various $p$ and $\theta$.**

figures for comparison reasons. Moreover, the input signal $x(n)$ is shown in all parts of Fig. 8 with dotted lines.

The results shown in the top panel of Fig. 8 demonstrate that the width of the detected edge is inversely proportional to $p$. On the other hand, the results shown in the bottom panel of Fig. 8 reveal the role of $\theta$ in emphasizing the rising edge, the falling edge, or both. It can be seen that our results are in complete accord with those reported in *Tseng & Lee (2015)*. It is also interested to see how close the outputs of our GFOD from those of the ideal one.

## CONCLUSION

This article has proposed a new design and implementation method for generalized fractional-order differentiators. The proposed method utilizes a composite structure of infinite impulse response subfilters whose parameters are optimized using a novel metaheuristic algorithm. The proposed metaheuristic algorithm is based on integrating whale optimization algorithm with a piecewise linear chaotic mapping and an adaptive inertia weight based on the hyperbolic tangent function. The superiority of the proposed metaheuristic over well-known benchmarks in designing accurate generalized fractional-order differentiators has been verified using average performance case and statistical studies. Compared to original whale optimizan algorthim, the new metaheuristic increases population diversity and promotes a faster convergence to optimal solutions, achieving better design performance with negligible computational burden. Moreover, the proposed generalized fractional-order differentiators have been compared with state-of-the-art designs. The results show that our designs can save a considerable computational complexity, while being competitive in terms of accuracy. An application example of signal edge detection has also been given to illustrate the effectiveness of our designs.

Future work can be conducted to investigate the effectiveness of the proposed design method in other applications, such as edge detection in images and the generation of secure single sideband signals for communication.

### Funding

This work was supported by the National Natural Science Foundation of China under Grants U24A20247 and 62272152, the Key R&D Program of Hunan Province under Grant 2024AQ2032, the Cultivation Project of Yuelu Mountain Industrial Innovation Center under Grant 2023YCII0123, the Shenzhen S&T Program under Grants JCYJ20220530160408019 and JCYJ20240813162405007, and the Guangdong Basic and Applied Basic Research Foundation under Grant 2023A1515011915. There was no additional external funding received for this study. The funders had no role in study design, data collection and analysis, decision to publish, or preparation of the manuscript.

### Grant Disclosures

The following grant information was disclosed by the authors:
National Natural Science Foundation of China: U24A20247 and 62272152.
Key R&D Program of Hunan Province: 2024AQ2032.
Yuelu Mountain Industrial Innovation Center: 2023YCII0123.
Shenzhen S&T Program: JCYJ20220530160408019 and JCYJ20240813162405007.
Guangdong Basic and Applied Basic Research Foundation: 2023A1515011915.

### Competing Interests

The authors declare that they have no competing interests.

### Author Contributions

- Mohammed Ali Mohammed Moqbel conceived and designed the experiments, performed the experiments, analyzed the data, performed the computation work, prepared figures and/or tables, authored or reviewed drafts of the article, and approved the final draft.
- Talal Ahmed Ali Ali conceived and designed the experiments, performed the computation work, authored or reviewed drafts of the article, and approved the final draft.
- Zhu Xiao analyzed the data, authored or reviewed drafts of the article, and approved the final draft.
- Amani Ali Ahmed Ali analyzed the data, performed the computation work, prepared figures and/or tables, and approved the final draft.

### Data Availability

The data and code are available in the Supplemental Files.

### Supplemental Information

Supplemental information for this article can be found online at http://dx.doi.org/10.7717/peerj-cs.2971#supplemental-information.

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
