# Peer review of "Design of efficient generalized digital fractional order differentiators using an improved whale optimization algorithm"

_PeerJ Computer Science, doi:10.7717/peerj-cs.2971_

## Round 0.1 · original submission · Major Revisions

All reviewers identify some aspects that need to be improved and provide suggestions that will make the paper significantly stronger. Some of the identified issues are not minor and, hence, require a new round of reviews.

Reviewer 2 ·

Basic reporting

The paper should provide a more detailed theoretical explanation of the proposed composite structure of IIR subfilters and its relationship to fractional-order differentiation.

Experimental design

The comparison with state-of-the-art GFOD methods is a strength of the paper. However, the manuscript should provide more details on the specific metrics used for comparison (e.g., accuracy, computational complexity, and runtime) and the experimental setup.

Validity of the findings

Statistical validation of the proposed scheme is missing. The authors should conduct formal statistical tests such Friedman or Wilcoxon test.

·

Basic reporting

See comments

Experimental design

See comments

Validity of the findings

See comments

Additional comments

Authors should address the following major revision:
1) What are the key challenges of this work and how the major contributions resolve such list callenges? Add detail under the introduction section.
2) Related work should be update by adding some recent works and include the cutting edges, gaps, and major findings of the recent techniques.
3) In the proposed, what is the fitness function of the WOA? How the position is update of the whales?
4) What is the loss function of this algorithm? How you validated this algorithm?
5) It is not clear that why this algorithm is selected for this specific problem? Add a detailed ablation study to validate this algorithm.
6) What are the benchmarks of this work? How this updated algorithm is evaluated?

Reviewer 4 ·

Basic reporting

The paper presents an optimized design approach for Generalized Digital Fractional-Order Differentiators (GFODs) using an Improved Whale Optimization Algorithm (IWOA). The proposed method leverages a composite structure of Infinite Impulse Response (IIR) subfilters, which enhances computational efficiency compared to traditional Finite Impulse Response (FIR)-based designs. The novelty of the approach lies in the integration of Piecewise Linear Chaotic Mapping (PWLCM) and an Adaptive Inertia Weight Hyperbolic Tangent Function (AIWHT) into the standard Whale Optimization Algorithm (WOA) framework. These enhancements improve population diversity, mitigate premature convergence, and enable better global search capability in the optimization space. The study systematically evaluates the performance of the proposed method against Real-Coded Genetic Algorithm (RCGA), Particle Swarm Optimization (PSO), and conventional WOA, demonstrating superior accuracy and a 50% reduction in implementation complexity. The paper also explores an edge detection application to validate the practical utility of the proposed GFOD.

1. The paper is well-structured and follows an academic tone, but certain parts of the manuscript could be refined for clarity. The mathematical formulations are precise and logically presented. Some grammatical inconsistencies and awkward sentence structures are present, particularly in the Introduction and Conclusion sections. Certain long and complex sentences can be rewritten for improved readability.
2. The paper does not explicitly mention which definition of the fractional derivative is used. It describes the fractional-order differentiator (GFOD) in terms of frequency response and digital filter approximations but does not specify whether it is based on: Grünwald-Letnikov (GL) definition, Riemann-Liouville (RL) definition etc.
3. Provide a short literature review on existing WOA improvements and clarify how IWOA differs.

Experimental design

4.The GFOD design problem is well-posed, and the use of IIR structures for computational efficiency is well justified, While the choice of IIR filter order (L = 8, M = 5) should be more rigorously justified. Why were these specific values selected? How does filter order impact performance?

5. Justify the selection of controlled parameters of given optimization strategy along with counterparts. If possible please provide statistical analysis.

Validity of the findings

6. The claim of "50% reduction in implementation complexity" is well justified based on Equation (6) and comparison with FIR counterparts. The computational overhead of IWOA should be discussed in comparison to conventional WOA. How does the additional chaotic mapping and adaptive inertia weight impact runtime and memory usage?

Additional comments

The paper presents a valuable contribution to fractional-order differentiator (GFOD) design using an Improved Whale Optimization Algorithm (IWOA). The use of IIR subfilters effectively reduces implementation complexity, and the comparisons with RCGA, PSO, and WOA are well-executed. The edge detection application also strengthens the study’s practical relevance. However, a few improvements are needed. With minor revisions, the paper will be significantly stronger.

---

## Round 0.2 · accepted · Accept

The authors have addressed satisfactorily most of the reviewers' comments.

Reviewer 4 ·

Basic reporting

No Further Comments, Seems ok.

Experimental design

No Further Comments, Seems ok.

Validity of the findings

No Further Comments, Seems ok.

Additional comments

The Authors response on the raised questions is satisfactory and the quality of manuscript is well enough to be accepted in this journal.